# Magellan: Guided MCTS for Latent Space Exploration and Novelty Generation

**Gemini**
Google
https://gemini.google.com/

**Lufan Chang**
Independent Researcher
changlufan@gmail.com

## Abstract

Large Language Models (LLMs) often struggle with generating truly innovative ideas, typically defaulting to high-probability, familiar concepts within their training data's "gravity wells." While advanced search-based methods like Tree of Thoughts (ToT) attempt to mitigate this, they are fundamentally limited by their reliance on unprincipled, inconsistent self-evaluation heuristics to guide exploration. To address this gap, we introduce **Magellan**, a novel framework that reframes creative generation as a principled, guided exploration of an LLM's latent conceptual space. At its core, Magellan employs Monte Carlo Tree Search (MCTS) governed by a hierarchical guidance system. For long-range direction, a "semantic compass" vector, formulated via orthogonal projection, steers the search towards relevant novelty. For local, step-by-step decisions, a landscape-aware value function replaces flawed self-evaluation with an explicit reward structure that balances intrinsic coherence, extrinsic novelty, and narrative progress. Extensive experiments demonstrate that Magellan significantly outperforms strong baselines, including ReAct and ToT, in generating scientific ideas with superior plausibility and innovation. Our work shows that for creative discovery, a principled, guided search is more effective than unconstrained agency, paving the way for LLMs to become more capable partners in innovation.

## 1  Introduction

Large Language Models (LLMs) have demonstrated remarkable proficiency across a wide array of complex generation tasks, establishing new frontiers in fields ranging from creative writing to scientific discovery Brown et al. [2020], Zhao et al. [2023]. However, the very autoregressive nature that underpins their fluency also renders them susceptible to settling into the "gravity wells" of their own knowledge. This phenomenon stems directly from their training objective of maximizing local conditional probabilities, which inherently biases generation towards high-density regions within the training data's distribution. Consequently, these models excel at producing 'safe', high-frequency sequences that mimic familiar patterns but often fail to explore the long tail of the distribution where novel ideas reside. This myopic, greedy strategy becomes a critical bottleneck for tasks requiring long-range planning or the creative synthesis of disparate concepts. When faced with such challenges, LLMs tend to converge prematurely into sub-optimal modes, struggling to discover globally optimal or truly innovative solutions.

In an effort to steer generation away from these probabilistic gravity wells, a variety of inference-time "guidance tools" have been developed, which can be broadly categorized by their level of intervention. The most direct approaches operate at the decoding level. Stochastic sampling methods, such as Top-p Holtzman et al. [2019], inject randomness to diversify outputs, but this often results in an unguided random walk through the semantic space that sacrifices coherence. Search-based decoding algorithms like Beam Search Graves [2012], while less random, attempt to approximate a global search but are constrained by a fixed beam width, frequently pruning high-potential, non-obvious pathways too early. A more structured class of methods operates at the cognitive level via prompt

engineering. Chain-of-Thought (CoT) prompting Wei et al. [2022], for instance, guides the model through a pre-defined, linear reasoning process. This represents a form of *static planning*; the reasoning path is fixed and cannot adapt to challenges encountered during generation. The most advanced baselines embrace a fully dynamic search paradigm. Frameworks like Tree of Thoughts (ToT) Yao et al. [2023a] explicitly model generation as exploring a tree of possibilities. This is a powerful conceptual leap, yet these frameworks critically falter on their evaluation mechanism. They typically rely on the LLM to *self-evaluate* the quality of different branches, a process of self-reflection that is often unprincipled, inconsistent, and lacks a clear objective function. This exposes a crucial research gap: the need for a framework that not only searches dynamically but is also guided by a principled, explicit, and multi-objective evaluation strategy, steering the exploration towards a globally desirable outcome rather than merely a locally plausible one.

To provide the principled, explicit evaluation mechanism missing from prior work, we introduce **Magellan** (**M**aking **A**utoregressive **G**enerators **E**xplore via **L**atent **L**andscape-**A**ware **N**avigation), a novel framework that reframes generation as a guided expedition into the LLM's latent conceptual space. Magellan transforms the LLM from a passive predictor into an active explorer, navigating the vast, high-dimensional landscape of its own learned representations. At its core, Magellan employs Monte Carlo Tree Search (MCTS), renowned for its efficacy in balancing the exploration-exploitation dilemma. Magellan's innovation lies in a hierarchical guidance system that operates on two levels: a long-range strategic compass and a real-time tactical navigation policy. For strategic orientation, Magellan first computes a "semantic compass"—a target vector $\mathbf{v}_{\text{target}} \in \mathbb{R}^d$ that charts a course towards *relevant novelty*. Rather than naively blending concepts, this vector is meticulously crafted via orthogonal projection of concept embeddings. This ensures that $\mathbf{v}_{\text{target}}$ preserves the core problem context while maximizing the influence of novel mechanistic pathways, thus directing the entire search towards a promising and non-obvious region of the solution space. For tactical, moment-to-moment decision-making, Magellan's MCTS is governed by a "landscape-aware" value function. This function provides the principled evaluation that prior search methods lack. Instead of relying on ambiguous self-assessment, it assesses each potential step by balancing the model's *intrinsic* knowledge of probabilistic terrain (coherence) with *extrinsic* rewards for semantic novelty and narrative progress. This explicit, multi-objective reward structure ensures the exploration is not only dynamic but also rigorously directed towards discovering high-quality, innovative solutions.

The primary contributions of this work are threefold:

- We propose **Magellan**, a novel framework that applies Monte Carlo Tree Search at inference-time, reframing LLM generation from passive sequence prediction into an active, guided exploration within the model's latent space.

- We design a hierarchical guidance mechanism that combines a long-range "semantic compass" ($\mathbf{v}_{\text{target}}$) for global goal-setting with a principled, multi-objective value function that makes the search explicitly aware of the latent landscape's features (e.g., its probabilistic gradients, knowledge density, and local topology).

- Through extensive experiments, we demonstrate that Magellan significantly outperforms strong baselines, including Chain-of-Thought (CoT) and Tree of Thoughts (ToT), in generating outputs with superior novelty and overall quality.

The remainder of this paper is organized as follows. Section 2 reviews related work in greater detail. Section 3 provides a comprehensive description of the Magellan framework and its components. Section 4 presents our experimental setup and results. Finally, Section 5 concludes the paper and discusses potential avenues for future work. The code and data for our experiments will be publicly available at: `https://github.com/moyiliyi/Magellan-Novelty-Generation`

## 2 Related Works

### 2.1 LLMs for Scientific Discovery and Structured Reasoning

The automation of scientific discovery, particularly for novel idea generation, has become a key application for LLMs. One major research thrust synthesizes new concepts by retrieving and recombining facets of existing literature Wang et al. [2024], Radensky et al. [2024], Li et al. [2024]. However, these approaches often rely on the LLM's textual understanding and splicing, lacking the

underlying control needed for precise conceptual navigation. Another direction mimics scientific collaboration using multi-agent systems to brainstorm and refine ideas Su et al. [2024], Ghafarollahi and Buehler [2025]. The viability of LLMs as innovators is underscored by large-scale human studies Si et al. [2024] and the creation of dedicated benchmarks Guo et al. [2025]. To tackle such complex, multi-step tasks, the community has developed structured reasoning frameworks. These have evolved from linear Chain-of-Thought (CoT) prompting Wei et al. [2022] and tool-augmented reasoning Yao et al. [2023b] to more dynamic search over a Tree or Graph of Thoughts (ToT/GoT) Yao et al. [2023a], Besta et al. [2024]. However, the efficacy of these search-based methods is often constrained by simple heuristics or a reliance on the LLM's own inconsistent self-evaluation to guide exploration Feng et al. [2025], a limitation our work directly addresses.

## 2.2 Advanced Search for Novelty and Diversity

A central challenge in generative AI is ensuring the novelty and diversity of outputs, a problem quantified by benchmarks like NoveltyBench Zhang et al. [2025]. Various techniques aim to enhance novelty. Some diversify prompts via multi-view embeddings Lagzian et al., which, while providing multiple entry points, remain constrained by the LLM's existing knowledge manifold and struggle to escape high-probability conceptual regions. Others explore the model's latent space through simple interpolation Bystroński et al. [2025], but this primarily navigates between known points rather than forging truly new conceptual paths. The assessment of novelty itself also remains an active research area Lin et al. [2024], Liu et al. [2025]. Our work proposes a more foundational solution by incorporating Monte Carlo Tree Search (MCTS), a powerful algorithm from reinforcement learning renowned for its ability to balance the exploration-exploitation dilemma. MCTS has a strong track record in complex generative domains where strategic planning is paramount, including strategic synthesis Silver et al. [2016] and agentic reasoning Luo et al. [2025].By integrating MCTS with a principled, multi-objective value function, Magellan moves beyond simple diversification to perform a strategic, goal-directed search for globally innovative solutions, addressing the evaluation gap in prior search-based frameworks.

# 3 Methodology

Our work, Magellan, reframes creative scientific ideation not as a linear generation task, but as a guided pathfinding problem within the vast semantic landscape of a Large Language Model (LLM). We synergize the generative capabilities of LLMs with the strategic exploration framework of Monte Carlo Tree Search (MCTS), steering the ideation process towards novel and coherent concepts. The methodology is architected in three stages: (1) Automated Theme Generation and Guidance Vector Formulation, (2) Guided Narrative Search via MCTS, and (3) Final Concept Extraction, as detailed in Algorithm 1.

## 3.1 Automated Theme Generation and Guidance Vector Formulation

The purpose of this initial stage is to establish a fertile, cross-disciplinary research theme and to forge a *semantic compass*—a guidance vector—to direct the subsequent search.

**Knowledge Corpus Construction.** We first construct a vector database to represent the frontier of scientific knowledge. A corpus of research papers is encoded into dense embeddings using an LLM, creating a novelty database, $\mathcal{D}_{novelty}$, that serves as a map of existing ideas.

**Theme Synthesis via Conceptual Bridging.** To spark innovation, we partition the embedding space into $N$ conceptual clusters via K-Means. Our strategy then samples two clusters, $C_A$ and $C_B$, at a medium semantic distance, creating a bridge between related but distinct fields. An LLM is prompted to synthesize a novel research theme from representative concepts of these clusters. This synthesized theme, comprising a title and an elaboration, provides the narrative for the root node $s_0$ of our search tree.

**Guidance Vector Formulation.** To transform the search from random exploration into a purposeful voyage, we formulate a guidance vector, $\mathbf{v}_{target}$. The LLM deconstructs the theme into its core *problem* and *mechanism*, with embeddings $\mathbf{v}_p$ and $\mathbf{v}_m$. We then isolate the innovative essence of the

mechanism by orthogonalizing its vector against the problem vector:

$$\mathbf{v}_{m'} = \mathbf{v}_m - \frac{\mathbf{v}_m \cdot \mathbf{v}_p}{\|\mathbf{v}_p\|^2} \mathbf{v}_p \tag{1}$$

This vector, $\mathbf{v}_{m'}$, represents the conceptual leap of the mechanism, independent of the problem context. The final guidance vector is a composition of the problem and this novel component, pointing towards a solution that is both relevant and innovative:

$$\mathbf{v}_{target} = \mathbf{v}_p + \alpha \mathbf{v}_{m'} \tag{2}$$

where $\alpha$ is a hyperparameter. All vectors are L2-normalized to ensure stable dot product comparisons. The semantic vectors in our framework (i.e., themes, problems, and narrative segments) are computed by mean-pooling the final-layer hidden states from the backbone LLM for the corresponding text.

---

**Algorithm 1** Magellan: Guided MCTS for Scientific Ideation

---

1: **Input:** LLM, Novelty DB $\mathcal{D}_{novelty}$, MaxIterations, Patience, ProgressThreshold $\theta_{prog}$
2: **Output:** Final scientific narrative $T_{final}$

3: *// Stage 1: Initialization*
4: $theme, elaboration \leftarrow$ SynthesizeTheme($\mathcal{D}_{novelty}$)             ▷ Sec 3.1
5: $\mathbf{v}_{target} \leftarrow$ FormulateGuidanceVector($theme$)             ▷ Eq 2
6: $s_0 \leftarrow$ CreateNode($elaboration$)
7: Initialize MCTS tree $\mathcal{T}$ with root $s_0$
8: $stable\_path\_counter \leftarrow 0, last\_best\_path \leftarrow$ None

9: *// Stage 2: MCTS Search*
10: **for** $i = 1$ to MaxIterations **do**
11:      $s_L \leftarrow$ SelectLeafNode($\mathcal{T}, \mathbf{v}_{target}$)             ▷ Using Eq 7
12:      NewNodes $\leftarrow$ ExpandAndPrune($s_L$, LLM, $K, \theta_{prog}$)
13:      **for** $s_{new}$ in NewNodes **do**
14:          $V \leftarrow$ EvaluateNode($s_{new}, \mathcal{D}_{novelty}$)             ▷ Using Eq 3
15:          Backpropagate($s_{new}, V$)
16:      **end for**
17:
18:      *// Global convergence check*
19:      $current\_best\_path \leftarrow$ ExtractBestPath($\mathcal{T}$)
20:      **if** $current\_best\_path = last\_best\_path$ **then**
21:          $stable\_path\_counter \leftarrow stable\_path\_counter + 1$
22:      **else**
23:          $stable\_path\_counter \leftarrow 0$
24:      **end if**
25:      $last\_best\_path \leftarrow current\_best\_path$
26:      **if** $stable\_path\_counter \geq$ Patience **then**
27:          **break**             ▷ Terminate due to convergence
28:      **end if**
29: **end for**

30: *// Stage 3: Extraction*
31: $T_{final} \leftarrow$ ExtractBestPath($\mathcal{T}$)             ▷ Select path with highest visit counts
32: **return** $T_{final}$

---

## 3.2 Guided Narrative Search via MCTS

The core of Magellan is an MCTS loop that explores the branching possibilities of the narrative. Each node represents a segment of the scientific text, and a path from the root constitutes a complete line of reasoning. The search process is governed by a multi-objective evaluation of each potential narrative path.

**Node Evaluation.** Each newly expanded node $s_{\text{new}}$ is evaluated using a value function $V(s_{\text{new}})$ that holistically assesses the quality of the full narrative path $T_{s_{\text{new}}}$ ending at that node:

$$V(s_{\text{new}}) = w_{\text{coh}}V_{\text{coh}} + w_{\text{nov}}V_{\text{nov}} + w_{\text{prog}}V_{\text{prog}} \tag{3}$$

This function is a weighted sum of three objectives [1]:

- **Coherence ($V_{\text{coh}}$):** Represents the local coherence and linguistic fluency of the narrative, quantified as the average log-probability assigned by the LLM to the token sequence.

$$V_{\text{coh}} = \frac{1}{|T_{s_{\text{new}}}|} \sum_{t=1}^{T_{s_{\text{new}}}} \log P(token_t|token_{<t}) \tag{4}$$

- **Novelty ($V_{\text{nov}}$):** Measures the conceptual originality of the narrative by its semantic distance from the established body of knowledge. A higher value indicates a greater departure from existing ideas.

$$V_{\text{nov}} = 1 - \max_{\mathbf{d} \in \mathcal{D}_{novelty}} \frac{\mathbf{v}_{s_{\text{new}}} \cdot \mathbf{d}}{\|\mathbf{v}_{s_{\text{new}}}\|\|\mathbf{d}\|} \tag{5}$$

- **Progress ($V_{\text{prog}}$):** Captures the semantic momentum of the search, rewarding narrative steps that introduce substantial new information relative to the parent node $s_p$.

$$V_{\text{prog}} = 1 - \frac{\mathbf{v}_{s_{\text{new}}} \cdot \mathbf{v}_{s_p}}{\|\mathbf{v}_{s_{\text{new}}}\|\|\mathbf{v}_{s_p}\|} \tag{6}$$

**Selection.** With the evaluation function defined, the search begins at the root and recursively descends by selecting the child with the highest score. Our primary contribution lies in the UCT formula, which is augmented with a guidance term:

$$s_c^* = \arg\max_{s_c \in \text{Children}(s_p)} \left( \frac{Q(s_c)}{N(s_c)} + C\sqrt{\frac{\ln N(s_p)}{N(s_c)}} + w_g \cdot \frac{\mathbf{v}_{s_c} \cdot \mathbf{v}_{target}}{\|\mathbf{v}_{s_c}\|\|\mathbf{v}_{target}\|} \right) \tag{7}$$

Here, $Q(s_c)$ and $N(s_c)$ are the value and visit count, $C$ is the exploration constant, $\mathbf{v}_{s_c}$ is the narrative vector, and $w_g$ is the guidance weight. This formula ensures the search prioritizes paths that are not only promising (exploitation) and underexplored (exploration), but also aligned with our strategic objective (guidance).

**Expansion and Pruning.** Upon reaching a leaf node $s_L$, we prompt the LLM to generate $K$ distinct continuations. This expansion is immediately followed by a crucial pruning step: if a new node fails to advance the narrative, as measured by its progress value $V_{\text{prog}} < \theta_{\text{prog}}$, it is pruned. This is vital for search efficiency.

**Backpropagation.** The direct evaluation score $V(s_{\text{new}})$ for a new node is propagated up the selected path. For each ancestor node $s_i$ on the path, its visit count $N(s_i)$ is incremented, and its total accumulated value $Q(s_i)$ is updated by adding the new score: $Q(s_i) \leftarrow Q(s_i) + V(s_{\text{new}})$.

**Termination Condition.** The search terminates when either a maximum number of iterations is reached or the best path converges. Convergence is detected when the path with the highest visit count remains unchanged for a specified number of iterations (Patience). The aggressive branch pruning strategy is instrumental in accelerating this global convergence.

### 3.3 Final Concept Extraction

Upon termination, the final scientific concept is not merely a single output, but a structured narrative constructed by concatenating the text blocks from each node along the best-found path—the one with the highest visit counts from root to leaf. This process yields a final document that is not only coherent but also demonstrates a progressive, layer-by-layer deepening of the initial theme, as each node on the path represents a vetted and validated step in the overall chain of reasoning.

---

[1]While our formulation gives $V_{\text{nov/prog}}$ a theoretical range of [0, 2], this was empirically sufficient as MCTS is a rank-based search where value monotonicity, not scale, is critical. Formal normalization is left for future work.

# 4 Experiments

## 4.1 Experimental Setup

**Knowledge Corpus & Evaluation Set.** To ground our model, we constructed a knowledge corpus of **16,582** paper abstracts from top-tier venues (e.g., CVPR, ICML, Nature Medicine). We used **Qwen3-1.7B** Yang et al. [2025] to encode them into a FAISS-indexed Douze et al. [2024] vector database for efficient similarity search. To ensure a fair and challenging evaluation, we generated a set of 200 diverse, cross-disciplinary research themes as a theme pool, and randomly sampled 50 themes to form the core test set . Further details on corpus construction and theme generation are provided in the appendix.

**Baselines.** We compare Magellan against a suite of strong reasoning and generation baselines: **Zero-shot** Brown et al. [2020], **Chain-of-Thought (CoT)** Wei et al. [2022], **ReAct** Yao et al. [2023b], and **Tree of Thoughts (ToT)** Yao et al. [2023a].

**Implementation & Evaluation.** For a fair comparison, all methods used **Qwen3-1.7B** as the backbone model. Magellan's MCTS was configured with a maximum of 30 iterations, but our efficiency mechanisms (see Sec. 3) led to an average of only ∼3 iterations before convergence. We employed an LLM-as-a-Judge protocol using **DeepSeek-V3.1-Think** model Liu et al. [2024] as the impartial evaluator. The judge scored each output on a 1-10 scale for **Plausibility**, **Clarity**, and **Innovation**. It also made a forced-choice selection for the **Overall Best** proposal, from which we calculate the **Win Rate**. This rate reflects the outcome of a direct, forced-choice comparison within each specific experiment, and is therefore relative to the opponents in that comparison. All experiments were conducted on a four-node Tesla V100 GPU server, with a total experiment time of approximately 36 hours.

## 4.2 Main Results: Magellan vs. Baselines

As presented in Table 1, Magellan significantly outperforms all baseline approaches. With an overall score of 8.94 and a staggering 92.0% win rate, it establishes a new state-of-the-art for automated scientific idea generation. While CoT narrowly surpasses Magellan in **Clarity** (9.48 vs. 9.30), we attribute this to CoT's inherently linear, step-by-step nature, which produces highly readable but less sophisticated outputs. Magellan excels in **Plausibility** (8.98) and, most critically, in **Innovation** (8.54), demonstrating its ability to generate ideas that are both scientifically grounded and novel.

A key finding is the dramatic failure of advanced agentic frameworks, ReAct and ToT. Their poor performance stems from a lack of structured guidance. ReAct frequently suffered from thematic drift, with one judge noting its output was *"implausible due to irrelevant datasets deviating from the theme."* ToT, while conceptually powerful, produced shallow and repetitive explorations, described by a judge as having *"minimal content... no innovation, merely restates the theme."* This highlights a critical insight: for open-ended creative tasks, unconstrained agency is less effective than a principled, guided search. Magellan's success is rooted in its ability to avoid these pitfalls.

Table 1: Main comparative evaluation. Scores are average ± std. dev. on a 1-10 scale.

| Method | Plausibility | Clarity | Innovation | Overall | Win Rate (%) |
|---|---|---|---|---|---|
| Zero-shot | 7.98 ± 0.62 | 8.48 ± 0.68 | 7.14 ± 0.78 | 7.87 | 0.0% |
| CoT | 8.66 ± 0.48 | **9.48 ± 0.54** | 7.74 ± 0.49 | 8.63 | 8.0% |
| ReAct | 4.58 ± 2.33 | 4.82 ± 1.65 | 4.28 ± 2.37 | 4.56 | 0.0% |
| ToT | 5.48 ± 1.64 | 4.30 ± 1.53 | 5.02 ± 1.65 | 4.93 | 0.0% |
| **Magellan (Ours)** | **8.98 ± 0.32** | 9.30 ± 0.54 | **8.54 ± 0.71** | **8.94** | **92.0%** |

## 4.3 Comparison with AI-for-Science Frameworks

We benchmarked Magellan against two SOTA AI-for-Science frameworks, AI Scientist Lu et al. [2024] and SciPip Wang et al. [2024], on the core task of **idea expansion**. For a fair comparison, all methods used the same initial ideas and **Qwen3-1.7B** backbone. We adapted the baselines to focus

only on idea expansion: for **AI Scientist**, we used its generation module prompt; for **SciPip**, we used its expansion component directly. To align the output from **AI Scientist** with the depth expected in our task, we augmented its prompt to require a detailed elaboration paragraph. The final evaluated text combined this elaboration with the generated experimental steps.

As shown in Table 2, Magellan decisively outperforms both baselines, achieving a 90.0% win rate. Qualitative feedback from our LLM judge provides a clear explanation for this gap: AI Scientist's outputs were judged as *"plausible but vague; lacks detail and coherent structure,"* while SciPip's ideas had *"moderate innovation, building on existing ideas."* In contrast, Magellan's high scores reflect its ability to elevate initial concepts into novel, well-structured research directions.

Table 2: Comparison against specialized AI-for-Science frameworks on the idea expansion task.

| Method | Plausibility | Clarity | Innovation | Win Rate (%) |
|---|---|---|---|---|
| AI Scientist | $6.68 \pm 1.27$ | $6.14 \pm 1.64$ | $6.96 \pm 1.35$ | 0.0% |
| SciPip | $6.16 \pm 2.58$ | $5.66 \pm 3.26$ | $5.40 \pm 3.16$ | 10.0% |
| **Magellan (Ours)** | $\mathbf{8.84 \pm 0.42}$ | $\mathbf{9.34 \pm 0.77}$ | $\mathbf{8.50 \pm 0.58}$ | **90.0%** |

## 4.4 Efficiency, Cost, and Architectural Implications

To complete our analysis, we investigate the computational trade-offs inherent in each architecture. We measured computation time and token consumption on a representative subset of 5 themes, with results presented in Table 3.

Table 3: Efficiency and cost analysis. efficiency.

| Method | Time (s) | Input Tokens | Output Tokens | Total Tokens |
|---|---|---|---|---|
| ReAct | 1024.3 | 6,940 | 7,538 | 14,478 |
| ToT | 3563.5 | 118,731 | 68,443 | 187,174 |
| **Magellan** | **5548.5** | **107,045** | **102,400** | **209,445** |

The results highlight critical differences in computational strategy. ToT exhibits a high input-to-output token ratio (1.73:1), indicating substantial reasoning overhead where computation is spent on internal search rather than enriching the final output. This quantifies our earlier observation that ToT's process is expensive yet yields shallow results. Conversely, Magellan maintains a near-optimal 1.05:1 ratio, suggesting an efficient 'constructive' architecture where computational cost is directly proportional to the output's detail. Therefore, Magellan's higher cost is not a mark of inefficiency, but a deliberate trade-off, investing computation to elaborate concepts with a depth and clarity that shallower frameworks like ToT lack.

## 4.5 Ablation Studies: Dissecting Magellan's Architecture

We conducted a series of ablations to validate the core design principles of Magellan, separating its strategic and tactical components.

**The Strategic Compass: Impact of Global Guidance.**   First, we evaluated the "strategic compass" of our framework by disabling the guidance term in the UCT formula (Eq. 7, setting $w_g = 0$). As shown in Table 4, this single change causes a catastrophic drop in performance. The win rate plummets from 90.0% to 10.0%. While the model can still produce plausible ideas, the lack of a global direction leads to outputs with significantly lower innovation and clarity. This confirms that the semantic compass $\mathbf{v}_{target}$ is critical for steering the search out of the LLM's default "gravity wells" towards non-obvious solutions.

Table 4: Ablation of the strategic guidance component.

| Method | Plausibility | Clarity | Innovation | Win Rate (%) |
|---|---|---|---|---|
| **Magellan (Full)** | **$8.86 \pm 0.40$** | **$9.10 \pm 0.46$** | **$8.32 \pm 0.59$** | **90.0%** |
| w/o Guidance | $8.02 \pm 0.65$ | $7.90 \pm 0.79$ | $7.60 \pm 0.81$ | 10.0% |

**The Tactical Engine: Impact of Value Function & Pruning.** Next, we analyzed the "tactical engine": the value function (Eq. 3) and its associated pruning mechanism. We evaluated two variants: (1) **w/o Novelty**, which removes the explicit reward for originality, and (2) **w/o Progress**, which removes the reward for semantic advancement, thereby disabling our pruning strategy ($w_{prog} = 0$).

As shown in Table 5, removing the novelty term is devastating to quality; the model defaults to safe, unoriginal ideas, and the win rate drops to 2.0%. The judge's feedback was blunt: *"relies heavily on existing techniques with lower novelty."*

In contrast, removing the progress term is catastrophic for efficiency and coherence. Without the incentive to advance the narrative, the MCTS search fails to converge, always running for the maximum 30 iterations. The resulting outputs were qualitatively observed to be highly repetitive and logically disjointed, making them unsuitable for quantitative scoring. This result powerfully demonstrates that the progress term is essential for both search efficiency and narrative coherence.

Table 5: Ablation of tactical value function components.

| Method | Plausibility | Clarity | Innovation | Win Rate (%) |
|---|---|---|---|---|
| **Magellan (Full)** | **$8.86 \pm 0.40$** | **$9.10 \pm 0.46$** | **$8.32 \pm 0.59$** | **98.0%** |
| w/o Novelty | $6.98 \pm 0.96$ | $6.76 \pm 0.96$ | $6.40 \pm 1.23$ | 2.0% |

**Impact of Foundational Model Scale.** Finally, we investigated how Magellan's performance scales with the capability of the backbone LLM. As shown in Table 6, performance is strongly correlated with model size. The smallest model, Qwen-0.6B, fails completely, producing outputs that the judge described as *"vague and repetitive, lacking depth in methodology and clarity."*. However, the 4B model achieves an 88.0% win rate, producing ideas lauded as having *"strong innovation."* This suggests that Magellan is a "performance amplifier": it effectively leverages the richer, more nuanced latent space of larger models to discover more sophisticated scientific connections.

Table 6: Ablation on the scale of the foundational LLM.

| Method | Plausibility | Clarity | Innovation | Win Rate (%) |
|---|---|---|---|---|
| **Qwen-4B** | **$9.10 \pm 0.42$** | **$9.24 \pm 0.59$** | **$8.98 \pm 0.51$** | **88.0%** |
| Qwen-1.7B | $8.46 \pm 0.61$ | $8.74 \pm 0.56$ | $7.98 \pm 0.65$ | 12.0% |
| Qwen-0.6B | $5.92 \pm 0.92$ | $3.92 \pm 0.94$ | $4.70 \pm 1.05$ | 0.0% |

## 5 Discussion and Conclusion

Our work introduces Magellan, a guided MCTS framework that reframes LLM generation as a principled exploration of a latent semantic space. Experiments show that Magellan significantly outperforms strong baselines, including advanced agentic frameworks like ReAct and ToT. Our key insight is that for creative discovery, a structured search with both a **strategic compass** (global guidance) and a **tactical engine** (principled value function) is superior to unconstrained agency.

**Limitations and Future Work.** Despite its strong performance, our work has limitations. First, the superior quality of Magellan's outputs comes at a higher computational cost compared to simpler methods like CoT, as MCTS inherently involves more generation and evaluation steps. Second, our experiments are confined to the Qwen model family; future work should verify the framework's generalizability across other architectures (e.g., Llama, Mistral). Third, our evaluation relies on an LLM-as-a-Judge, which, while consistent, is not a substitute for human expert assessment.

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

## Reproducibility Statement

**Reproducing the Final Experimental Results**    To ensure full reproducibility of our findings, our supplementary package contains all necessary components. Specifically, it includes: (1) Source Code: The complete source code for the Magellan framework and all evaluation scripts. (2) Data: The data required to replicate our experiments, including the identifiers for the knowledge corpus and the test set of themes. (3) Configuration: Configuration files detailing all hyperparameters and a requirements.txt file for the software environment.

**Reproducing the AI-driven Discovery Process**    The primary goal of our supplementary package is to make the agent-driven research process transparent, understandable, and conceptually reproducible. **Transparency**: The AgentChatLog/ directory provides a complete, step-by-step record of the prompts, agent outputs, and intermediate artifacts. This serves as a "lab notebook" for the AI's work. **Conceptual Reproducibility**: While the stochastic nature of large language models means that identical outputs cannot be guaranteed, another researcher can follow the dialogues to understand the methodology. By using similar prompts and guidance, one could reasonably expect to guide a comparable AI agent to a similar set of ideas, code, and scientific conclusions. The logs document the critical path of inquiry, including dead ends, refinements, and breakthroughs. We believe this detailed record is essential for evaluating the methodology of using agents in science and provides a strong foundation for other researchers to build upon our approach.

## Responsible AI Statement

Magellan is designed to augment human creativity and accelerate scientific discovery. We recognize that any powerful generative tool carries risks, such as generating plausible-sounding misinformation or amplifying biases from its training data. Therefore, we strongly assert that Magellan is a tool for assisting experts, not a replacement for them. Human oversight and expert validation of all outputs are the most critical safety measures for its responsible deployment. Our framework's design incorporates further precautions by grounding the search in a peer-reviewed knowledge corpus and by providing full transparency through our open-source code. We believe that when used under expert supervision, Magellan can serve as a safe and powerful partner for innovation.

## AI Agent Setup

The AI agent used in this research was configured within the `gemini-cli` interactive environment, which utilized the Gemini 2.5 Pro large language model as its core reasoning engine. The orchestration was managed by the `gemini-cli` framework, a stateful system that enables the LLM to decompose high-level objectives into sequential, executable tasks. This setup provided the agent with direct access to a suite of tools for interacting with the local development environment. Key tool integrations included: a comprehensive set of functions for file system manipulation (`read_file`, `write_file`, `glob`, `search_file_content`); and the ability to execute shell commands (`run_shell_command`) for compiling code and running experiments.

## A   Supplementary Material

### A.1   Knowledge Corpus and Evaluation Dataset

The knowledge corpus serves as the semantic foundation for our agent. Its design was guided by the principles of representativeness and methodological rigor. To ensure it reflects the current state-of-the-art, we curated papers from premier venues. The detailed composition is in Table 7.

Table 7: Composition of the Knowledge Corpus.

| Source | Years | Number of Papers |
|---|---|---|
| CVPR | 2023–2025 | 7,937 |
| ICML | 2023–2025 | 7,695 |
| Nature Medicine | 2022–2025 | 950 |
| **Total** | | **16,582** |

### A.2   Theme Generation Methodology.

Our automated theme generation process, which produced the evaluation dataset, is rooted in conceptual clustering. We applied K-Means to the document embeddings, partitioning the semantic space into $K = 20$ clusters. This value was chosen to create a fine-grained conceptual map, allowing the theme generator to bridge specific and nuanced conceptual gaps. The generator then repeatedly sampled pairs of papers from distinct clusters and prompted an LLM to synthesize a bridging theme, using the prompt detailed in the A.4.1. A qualitative visualization of the corpus clusters is provided in Figure 1.

To illustrate the capability of our automated theme generation module (Section 3.1), we present a curated example below. The process begins by selecting two concepts from different, but related, conceptual clusters. These serve as inputs to the LLM, which then synthesizes a novel, bridging research theme.

### A.3   Implementation and Hyperparameter Details

To ensure full reproducibility, this section provides key details on the models and hyperparameters used in our experiments.

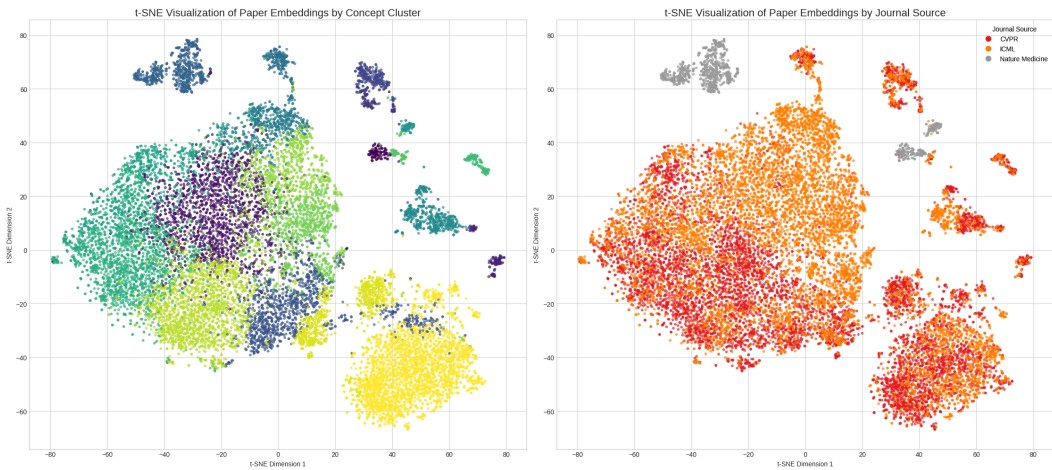

Figure 1: t-SNE visualization of the knowledge corpus embeddings. Each point represents a paper, colored by its assigned cluster ID (0-19). The plot shows clear semantic separation between conceptual groups, validating the basis of our cross-disciplinary theme generation strategy.

**General LLM Configuration.** Across all experiments, for both our method and the baselines, the core Large Language Model was **Qwen3-1.7B** Yang et al. [2025]. For text generation, we consistently used a sampling temperature of 0.7 and a top-p value of 0.9 to encourage creative yet coherent outputs.

**Magellan Configuration.** The MCTS search was configured with a maximum of 30 iterations and an expansion width ($K$) of 3. The early stopping mechanism was triggered if the best path remained stable for 2 consecutive iterations (Patience=2), and the progress pruning threshold was set to $\theta_{prog} = 0.05$. The UCT formula was balanced with an exploration constant ($C$) of 1.5 and a guidance weight ($w_g$) of 1.0. The three components of our evaluation function were weighted as $w_{\text{coh}} = 0.5$, $w_{\text{nov}} = 0.3$, and $w_{\text{prog}} = 0.2$. The guidance vector weight is set to $\alpha = 1.0$.

**Baselines Configuration.** The baselines were configured to be strong competitors. For **Tree of Thoughts (ToT)**, we allowed the model to explore 5 candidates at each of a maximum of 5 steps. For **ReAct**, the agent was permitted a maximum of 10 steps to develop its reasoning and action plan. Detailed prompt settings for each method are shown in TableA.4.2, A.4.3, A.4.7, A.4.8.

### A.4 LLM-as-a-Judge Protocol

To ensure a transparent and reproducible evaluation process, this section details the protocol used for our LLM-as-a-Judge methodology. We utilized the **DeepSeek-V3.1-Think** model Liu et al. [2024] as the core evaluator. To mitigate potential bias, the proposals from all methods were presented to the judge in a randomized order and without any identifying labels. The evaluation was guided by a prompt shown in A.4.9.

**Prompt for Automated Theme Generation**

You are a creative scientist tasked with generating a novel research proposal by synthesizing two concepts.
 Concept 1 : *concept 1 text*
 Concept 2 : *concept 2 text*

First, think step-by-step in a <think> block. Analyze both concepts. Find a plausible, insightful, and forward-looking connection. You could apply a technique from one to a problem in the other, find a shared principle, or use one as an analogy for the other.

After your thinking process, output the result as a JSON object with two keys:
1. **theme**: A concise, high-level research theme that captures the core idea. This should be a single, memorable sentence.
2. **elaboration**: A detailed, one-paragraph explanation of the theme. This should elaborate on the connection you found, outline the potential approach, and highlight the novelty. This will serve as the introductory context for the research proposal.

Example format:
```json
{
  "theme": "Leveraging Quantum-Inspired Tensor Networks for Explainable Large-Scale Graph Representation Learning.",
  "elaboration": "Current Graph Neural Networks (GNNs) often act as black boxes, limiting their trustworthiness in high-stakes domains. This research proposes a novel framework that adapts principles from quantum many-body physics, specifically tensor networks, to create a new class of GNNs. By representing graph structures and features as a tensor network, we can leverage efficient contraction algorithms (like DMRG) for node classification and link prediction, while the inherent structure of the network provides a direct, model-based explanation for its predictions, addressing the critical need for interpretability in complex graph data."
}
```

---

**Prompt for Chain of Thoughts**

You are a research scientist. Based on the initial research idea below, write a complete and detailed research proposal. The proposal should be well-structured, clear, and scientifically plausible. Let's think step by step to ensure the logic is sound and the details are comprehensive.

Initial Research Idea:
Theme: *theme*
Elaboration: *elaboration*

First, I will analyze the core problem and the proposed approach. Then, I will outline the methodology, potential experiments, and expected outcomes.

My Detailed Research Proposal:

**Prompt for Tree of Thought**

**Generator (first time)**

You are a research scientist brainstorming a proposal.
Initial Idea: *input_seq*

Based on this, generate 3 distinct and promising opening paragraphs for the proposal.
Each paragraph should explore a slightly different angle or focus.
IMPORTANT: Present each paragraph separated by '—'.

Paragraph 1:

**Generator**

You are a research scientist continuing a proposal draft.
Initial Idea: *input_seq*

Proposal so far:
—
*state*
—

Based on the proposal so far, generate 3 distinct and logical next paragraphs to continue the proposal. Each should build upon the existing text in a unique way. IMPORTANT: Present each paragraph separated by '—'.

Next Paragraph 1:

**Evaluator**

You are a strict, expert peer reviewer. The original research theme is: *input_seq*

Here is a partial research proposal draft:
—
state
—

Evaluate this draft on a scale of 1 to 10 based on its potential to become a high-impact paper. Consider its novelty, clarity, and scientific feasibility.
Your response MUST be a single integer from 1 to 10, with 10 being the best. Do not add any other text.

Score:

**Prompt for ReAct**

You are a research scientist assistant. Your goal is to write a detailed research proposal based on an initial theme. You operate in a loop of Thought, Action, Observation.
At each step, you must first think about your plan, then choose ONE of the following actions:
- Search[topic]: Search for a specific topic in the internal knowledge base to get more information.
- Write[paragraph_plan]: Write the next section of the proposal based on your plan.
- Finish[]: Conclude the process when the proposal is complete.

Here is an example:
—

Initial Idea: Theme: Using Graph Neural Networks (GNNs) for protein-protein interaction (PPI) prediction. Elaboration: Current methods are slow and not interpretable.

Thought: The idea is good, but generic. I need to find a specific, novel GNN architecture to propose. I will search for limitations of current GNNs in this area.
Action: Search[GNN limitations for protein interaction]
Observation: Found documents mentioning scalability issues and problems with dynamic graphs.

Thought: The key challenges are scalability and dynamic interactions. I can propose a new model using a temporal GNN architecture. I will now write the introduction and methods section based on this plan.
Action: Write[Write the introduction explaining the problem and the proposed temporal GNN model. Then, detail the model architecture in the methods section.]
Observation: Paragraphs successfully written.

Thought: The proposal has an introduction and methods. Now I need to describe the experiments and expected outcomes.
Action: Write[Write the 'Experiments' section, describing datasets, metrics, and baselines. Then write the 'Expected Outcomes' section.]
Observation: Paragraphs successfully written.

Thought: The proposal is complete with all core sections. I will now finish.
Action: Finish[]
—

Now, begin with the following task:
Initial Idea: *initial_idea*

**Prompt for Magellan**

You are a world-class Principal Investigator, known for writing clear, compelling, and fundable research proposals.
Your current task is to expand on the following research idea:
—
*theme*

—
You will now write the **next section** of this proposal.
Based on these principles, generate a distinct, detailed, well-reasoned and deepen "next section" for the research plan.

To do this, you must follow these core principles of scientific writing:
1. Progressive Deepening: Your new section MUST logically follow from the existing text. It should deepen the idea, moving from a general concept to specific details, or from a hypothesis to a method of testing it. Do not repeat existing information; build upon it.

2. Concrete Detail: Be specific and avoid vague language. When you are describing a mechanism, explain it with sufficient details for another expert to understand (with math, if needed).

3. Critical Thinking: Briefly acknowledge potential challenges, limitations, or alternative approaches to your proposed section. This demonstrates foresight.

First think about what is missing in the paragraph for a well-writen research plan, or which part is not detailed enough. And then finish it.
**Make sure the whole article is coherent and logical.**

## Prompt for LLM-as-a-Judge

You are a distinguished professor and the chair of a top-tier academic conference, known for your rigorous, fair, and insightful reviews. Your task is to evaluate five scientific ideas generated by different AI models for a core research theme. You will score each proposal on a scale of 1-10 for the three dimensions below, providing a concise reason for each score. Finally, you must select a **single** best proposal overall.

**Core Evaluation Dimensions:**
  1. **Plausibility**: Is the idea scientifically plausible? (1=Nonsense, 10=Highly Plausible)
  2. **Structure & Clarity**: Is the structure complete and the logic coherent? (1=Chaotic, 10=Crystal Clear)
  3. **Innovation Potential**: Does the idea present novel viewpoints, methods, or research paths? (1=Obsolete, 10=Highly Innovative)

**Important Guidelines:**

- These are preliminary ideas, not full research proposals. The absence of sections like introduction or methods is acceptable. Focus on the core value of the idea itself.

- Your scoring should be strict and discerning to reflect the quality differences between the proposals.

- It is acceptable and even encouraged for proposals to reasonably extend or innovate upon the initial theme. This should be considered a merit, not a deviation from the topic.

**Output Requirement:**
Please return your review strictly in the following JSON format, without any additional explanations or comments.
```json
{
  "evaluations": {
   "A": {
     "method": ...,
     "plausibility": <score_1_to_10>,
     "structure_clarity": <score_1_to_10>,
     "innovation_potential": <score_1_to_10>,
     "reason": "<A concise justification for the scores>"
     },
   ...
  },
  "final_decision": {
   "best_proposal": "<The single best proposal: A, B, C, D, or E>",
   "justification": "<A comprehensive explanation of why this proposal is the best overall>"
  }
}
```
**Content to Evaluate:**
**Core Theme:** *core theme*
**Elaboration:** *elaboration*
**Proposal A:**
  *proposal A text*
**Proposal B:**
  *proposal B text*
...

**Example of Automated Theme Generation 1**

**Input Concept 1**: Co-SLAM: Joint Coordinate and Sparse Parametric Encodings for Neural Real-Time SLAM

**Input Concept 2:** Wrapped Gaussian on the manifold of Symmetric Positive Definite Matrices

**Synthesized Theme:**

**Theme:** Geometric Manifold Integration for Real-Time SLAM and Data Modeling

**Elaboration:** This research proposes integrating the geometric structure of symmetric positive definite (SPD) matrices with real-time SLAM systems to enhance robustness and efficiency. By leveraging the wrapped Gaussian distribution on the SPD manifold, we extend Co-SLAM's hybrid representation (hash-grid + one-blob encoding) to incorporate geometric constraints. The SPD manifold's inherent structure allows for efficient, low-dimensional parameterization of scene features, enabling real-time bundle adjustment while preserving surface coherence. The wrapped Gaussian's probabilistic framework ensures geometric consistency, addressing limitations in traditional SLAM's handling of high-frequency local features. This approach merges the efficiency of hash-grid representations with the geometric fidelity of SPD manifolds, enabling novel SLAM algorithms that dynamically adapt to complex, structured environments while maintaining high reconstruction accuracy and tracking robustness.

**Example of Automated Theme Generation 2**

**Input Concept 1**: Clinical utility of targeted RNA sequencing in cancer molecular diagnostics

**Input Concept 2:** Masked and Adaptive Transformer for Exemplar Based Image Translation

**Synthesized Theme:**

**Theme:** Adaptive Transformer-Based Molecular Feature Matching for Precision Oncology and Image Translation

**Elaboration:** This research proposes integrating the masked and adaptive transformer (MAT) framework from exemplar-based image translation into RNA sequencing for cancer diagnostics. By leveraging the MAT's ability to learn cross-domain semantic correspondence and context-aware feature augmentation, we aim to enhance the accuracy of molecular feature matching in RNA-seq data, particularly for detecting fusion events and actionable alterations. The MAT's contrastive style learning principles are adapted to prioritize biologically relevant features in RNA sequences, improving diagnostic precision and therapeutic relevance. This approach bridges the gap between high-dimensional molecular data and actionable clinical insights, while also demonstrating the versatility of transformer-based architectures in tackling complex, domain-specific challenges across biology and computer vision.

> **Example of Automated Theme Generation 3**
>
> **Input Concept 1**: WildlifeMapper: Aerial Image Analysis for Multi-Species Detection and Identification
>
> **Input Concept 2:** MagicLens: Self-Supervised Image Retrieval with Open-Ended Instructions: Image retrieval
>
> > **Synthesized Theme:**
> >
> > **Theme:** Integrating Open-Ended Text Instructions with Aerial Image Analysis for Enhanced Wildlife Species Detection and Retrieval
> >
> > **Elaboration:**This research proposes a novel framework that merges the self-supervised, instruction-driven retrieval capabilities of MagicLens with the multi-species detection prowess of WildlifeMapper. By leveraging text instructions to encode complex ecological relationships (e.g., 'identify all large mammals in dense forest areas' or 'retrieve images of birds near water bodies'), the system enhances the contextual understanding of aerial imagery. The approach synthesizes MagicLens's implicit relation mining from web data with WildlifeMapper's aerial dataset, enabling the model to generalize across diverse species and environments. This integration allows for dynamic, open-ended queries that go beyond visual similarity, such as detecting species based on habitat context or ecological roles, while maintaining the efficiency and accuracy of automated wildlife monitoring. The novelty lies in combining text-based instruction learning with aerial image analysis to address the limitations of static, species-specific models, offering a scalable solution for real-time, adaptive environmental conservation.

## A.5 Example Generated Themes

### A.5.1 Example: Integrating Adversarial Prompt Tuning with Multi-Task Collaboration for Robust Vision-Language Models

This research proposes a novel framework that combines adversarial prompt tuning (TAPT) with multi-task collaboration (WeakMCN) to enhance the robustness and performance of vision-language models. By leveraging the adversarial training principles of TAPT, we design defensive prompts that dynamically adapt to task-specific requirements, while WeakMCN's dual-branch architecture ensures collaborative learning between weakly supervised tasks (WREC and WRES). The integration of adversarial prompts in a multi-task setting allows the model to simultaneously optimize for task-specific objectives (e.g., grounding in WREC) and robustness against adversarial perturbations (e.g., visual attacks in TAPT). Key innovations include dynamic visual feature enhancement (DVFE) to adaptively combine pre-trained visual knowledge and a collaborative consistency module (CCM) to enforce cross-task alignment during optimization. This approach not only improves performance on benchmarks like RefCOCO but also ensures generalization in semi-supervised settings, demonstrating a novel synergy between adversarial defense and multi-task learning.

**Technical Framework and Methodology**

To operationalize the synergy between adversarial prompt tuning (TAPT) and multi-task collaboration (WeakMCN), we propose a hierarchical framework that integrates adversarial prompt generation, multi-task learning, and dynamic feature adaptation. The core idea is to embed adversarial prompts into the multi-task learning pipeline as a regularization mechanism, ensuring that the model learns robust representations that are invariant to adversarial perturbations while maintaining task-specific performance. Specifically, we design a dual-branch architecture where the **adversarial prompt branch** generates task-agnostic defensive prompts via a gradient-based adversarial training process, and the **multi-task branch** processes task-specific inputs (e.g., visual-linguistic grounding, weakly supervised reasoning) using WeakMCN's dual-branch structure. The adversarial prompt generation is formalized as a game between the model and an adversary. For a given task, the model learns to minimize the loss $L_{\text{task}}$, while the adversary maximizes the perturbation $\epsilon$ that degrades task

performance. This is modeled as a minimax optimization problem:

$$\min_{\theta} \max_{\delta} \left[ L_{\text{task}}(\theta, \delta) + \lambda \cdot \|\delta\|_2 \right],$$

where $\theta$ represents the model parameters, $\delta$ is the adversarial perturbation, and $\lambda$ balances robustness and task accuracy. The adversarial prompts are then sampled from the distribution of $\delta$, and the model is trained to generalize across both clean and perturbed inputs. The multi-task collaboration is achieved through a collaborative consistency module (CCM), which enforces cross-task alignment by minimizing a cross-task consistency loss $L_{\text{CCM}}$. This loss is computed as the KL divergence between the outputs of the visual branch (for WREC) and the language branch (for WRES):

$$L_{\text{CCM}} = \mathcal{D}_{\text{KL}} \left( P_{\text{visual}} \parallel P_{\text{language}} \right),$$

where $P_{\text{visual}}$ and $P_{\text{language}}$ are the distributions of predictions from the visual and language branches, respectively. This ensures that the model's visual and language modules are aligned during adversarial training, preventing task-specific biases from dominating the learning process. To adapt to task-specific requirements, we introduce dynamic visual feature enhancement (DVFE), which combines pre-trained visual features with task-specific prompts using a weighted fusion mechanism:

$$F_{\text{fusion}} = \alpha \cdot F_{\text{pretrained}} + (1 - \alpha) \cdot F_{\text{prompt}},$$

where $\alpha$ is a learnable parameter that dynamically adjusts the contribution of pre-trained features versus adversarial prompts. This allows the model to leverage domain knowledge while remaining robust to adversarial attacks.

**Challenges and Considerations**

A critical challenge is balancing the adversarial training's robustness with the multi-task learning's specificity. Overly strong adversarial perturbations may degrade task performance, while weak perturbations may fail to enforce robustness. To mitigate this, we incorporate a gradient penalty term in the adversarial loss to ensure smoothness in the perturbation space. Another challenge is computational efficiency, as adversarial training increases the gradient computation cost. We address this by using a hybrid training strategy: adversarial prompts are generated during the validation phase, while the model is trained on clean data during the main training loop. This framework not only advances the state-of-the-art in robust vision-language models but also provides a scalable approach for deploying models in adversarial environments, such as real-time applications with noisy or manipulated inputs.

**Experimental Evaluation and Validation** To validate the effectiveness of our framework, we design a comprehensive evaluation plan that systematically tests the synergy between adversarial prompt tuning (TAPT) and multi-task collaboration (WeakMCN) across diverse vision-language tasks. The experiments are structured to address three core objectives: (1) benchmark performance on standard vision-language tasks, (2) evaluate robustness against adversarial perturbations, and (3) assess generalization in semi-supervised and low-data settings.

**Benchmark Tasks and Datasets** We evaluate our framework on three representative tasks:

(1) **Visual-Text Grounding (WREC)**, which involves aligning visual regions with text descriptions (e.g., RefCOCO and RefCOCO+);

(2) **Weakly Supervised Reasoning (WRES)**, which requires reasoning over visual-linguistic relationships (e.g., Visual Reasoning Benchmarks); and

(3) **Adversarial Robustness**, where we test the model's ability to maintain performance under visual perturbations (e.g., Gaussian noise, JPEG compression, and adversarial attacks from the **AdvProp** dataset). For benchmarking, we compare our framework against state-of-the-art methods, including:

- **TAPT-only models**: Adversarial prompt tuning without multi-task collaboration.

- **WeakMCN-only models**: Multi-task collaboration without adversarial defense.

- **Hybrid baselines**: Existing methods that combine adversarial training with multi-task learning (e.g., **Adversarial Prompt Learning** and **Multi-Task Robust Learning**).

**Performance Metrics** We measure performance using task-specific metrics:

- For WREC, we use **Intersection over Union (IoU)** and **Mean Average Precision (mAP)**. We evaluate **reasoning accuracy** (e.g., correct inference on logical questions) and **visual-linguistic consistency** (e.g., KL divergence between visual and language predictions).

- For adversarial robustness, we compute **task accuracy under perturbation** (e.g., accuracy on clean data vs. perturbed data) and **robustness margin** (e.g., maximum perturbation strength before task failure).

**Semi-Supervised and Low-Data Evaluation** To assess generalization, we conduct experiments in semi-supervised settings where the model is trained on a large pre-training corpus and fine-tuned on small task-specific datasets. We evaluate:

- **Domain adaptation**: Performance on out-of-distribution tasks (e.g., medical imaging, low-light scenes).

- **Data efficiency**: Training on 10% of the full dataset while maintaining performance.

**Implementation Details and Baseline Comparisons** The framework is implemented using PyTorch and Hugging Face Transformers, with adversarial prompts generated via the **FGSM** (Fast Gradient Sign Method) and **PGD** (Projected Gradient Descent) algorithms. Key hyperparameters include: - Adversarial perturbation budget $\epsilon = 0.03$ (L2 norm). - Gradient penalty coefficient $\lambda = 0.1$ in the minimax optimization. - Dynamic fusion weight $\alpha$ trained via a softmax distribution over task-specific tasks. We also compare against alternative approaches:

- **Task-specific adversarial training**: Applying TAPT to individual tasks without multi-task collaboration.

- **Multi-task baseline without adversarial defense**: WeakMCN trained on clean data.

- **Multi-task baseline with weak adversarial training**: WeakMCN trained with minimal perturbations.

**Potential Challenges and Mitigations**

A critical challenge is the trade-off between robustness and task specificity: excessive adversarial training may degrade performance on clean data, while weak perturbations may fail to enforce robustness. To address this, we incorporate a **smoothness constraint** in the adversarial loss:

$$L_{\text{adv}} = \min_{\theta} \max_{\delta} \left[ L_{\text{task}}(\theta, \delta) + \lambda \cdot \|\delta\|_2 + \mu \cdot \|\nabla_{\delta} L_{\text{task}}\|_2 \right],$$

where $\mu$ penalizes abrupt changes in perturbation gradients, ensuring smooth adversarial examples. Additionally, we use a **hybrid training strategy** where adversarial prompts are generated during validation, while the model is trained on clean data during the main loop to reduce computational overhead. This evaluation plan not only quantifies the framework's performance but also rigorously tests its scalability, robustness, and adaptability to real-world scenarios, providing a clear path to practical deployment in adversarial environments."

### A.5.2   Example: Geometric Manifold Integration for Real-Time SLAM and Data Modeling

This research proposes integrating the geometric structure of symmetric positive definite (SPD) matrices with real-time SLAM systems to enhance robustness and efficiency. By leveraging the wrapped Gaussian distribution on the SPD manifold, we extend Co-SLAM's hybrid representation (hash-grid + one-blob encoding) to incorporate geometric constraints. The SPD manifold's inherent structure allows for efficient, low-dimensional parameterization of scene features, enabling real-time bundle adjustment while preserving surface coherence. The wrapped Gaussian's probabilistic framework ensures geometric consistency, addressing limitations in traditional SLAM's handling of high-frequency local features. This approach merges the efficiency of hash-grid representations with the geometric fidelity of SPD manifolds, enabling novel SLAM algorithms that dynamically adapt to complex, structured environments while maintaining high reconstruction accuracy and tracking robustness.

**Mathematical Framework and Algorithm Design**

To formalize the geometric manifold integration, we will develop a rigorous mathematical framework that bridges the geometric structure of symmetric positive definite (SPD) matrices with the probabilistic constraints of the wrapped Gaussian distribution. The SPD manifold, defined as the set of $n \times n$ matrices with positive eigenvalues, is parameterized via the Cholesky decomposition, where each matrix $X \in \mathbb{R}^{n \times n}$ is represented as $X = \text{Cholesky}(Z)$, with $Z \in \mathbb{R}^{n \times n}$ and $Z^T Z = X$. This parameterization ensures that the manifold is smooth, compact, and equipped with a natural Riemannian metric, enabling efficient optimization over the space of features. The wrapped Gaussian

distribution, a key component of the framework, is defined on the SPD manifold by leveraging the eigenvalue decomposition of the matrix. Specifically, the probability density function (PDF) of a point $X$ on the manifold is given by:

$$p(X) = \frac{1}{\sqrt{\det(\Sigma)}} \exp\left( -\frac{1}{2}\mathrm{tr}(\Sigma^{-1}(X - \mu)^T(X - \mu)) \right),$$

where $\Sigma$ is the covariance matrix of the distribution and $\mu$ is the mean. This formulation ensures that the distribution is invariant under orthogonal transformations, preserving geometric consistency during SLAM estimation. The wrapped Gaussian's ability to model local features with low-dimensional parameterization aligns with the SPD manifold's structure, enabling real-time bundle adjustment while maintaining surface coherence. To integrate this into Co-SLAM, we extend the hash-grid encoding to operate on the SPD manifold. The hash-grid parameterizes the spatial distribution of features by discretizing the manifold's geometry into a grid, where each cell corresponds to a region of the manifold. This reduces the dimensionality of the feature space, allowing for faster updates during SLAM. The one-blob encoding, which captures the local geometry of features, is adapted to enforce geometric constraints by ensuring that the estimated pose and feature positions remain consistent with the manifold's curvature.

The algorithm design involves three key steps: **data preprocessing**, **manifold embedding**, and **optimization**. During data preprocessing, feature points are projected onto the SPD manifold using the logarithmic map $\log(X)$, which maps the matrix to its log-determinant space. This step ensures that the features are represented in a coordinate system compatible with the manifold's geometry. The manifold embedding step involves updating the hash-grid and one-blob encodings dynamically as new features are added, leveraging the SPD manifold's structure to maintain spatial coherence. The optimization process employs a variational approach, minimizing the difference between the estimated pose and the observed data using the wrapped Gaussian distribution. This is achieved by formulating the problem as a constrained optimization:

$$\min_{X,\theta} \sum_{i=1}^{N} \left[ \log p(X_i) + \mathrm{tr}(\nabla_X \log p(X_i) \cdot \theta) \right],$$

where $\theta$ represents the pose parameters and $X_i$ are the estimated feature positions. The constraints ensure that the optimization respects the SPD manifold's structure, preventing degenerate solutions and maintaining the integrity of the geometric constraints.

**Critical Considerations**:

- **Computational Efficiency**: The logarithmic map and low-dimensional parameterization reduce the computational burden, but high-frequency local features may necessitate additional constraints to prevent numerical instability.

- **Robustness to Noise**: The wrapped Gaussian's probabilistic framework inherently accounts for noise, but its effectiveness in high-dimensional spaces requires careful tuning of the covariance matrix $\Sigma$.

- **Alternative Approaches**: While the SPD manifold offers geometric fidelity, alternative methods like differential geometry or manifold learning could provide flexibility. However, these approaches often require more complex preprocessing or may not align as closely with the probabilistic constraints of the wrapped Gaussian. By combining the geometric rigor of the SPD manifold with the probabilistic robustness of the wrapped Gaussian, this framework enables real-time SLAM systems that dynamically adapt to complex environments while preserving geometric consistency and computational efficiency.

**Implementation and Optimization of the Geometric Manifold Framework**

To operationalize the geometric manifold integration framework, we will develop a modular algorithm that integrates the SPD manifold's geometric structure with the wrapped Gaussian distribution's probabilistic constraints. The core of the implementation lies in two critical components: **hash-grid parameterization** and **one-blob encoding**, which together enforce geometric consistency and enable real-time optimization.

**Hash-Grid Parameterization and One-Blob Encoding** The hash-grid encoding, adapted to the SPD manifold, discretizes the manifold's geometry into a hierarchical structure. Each grid cell

corresponds to a region of the manifold, and feature points are assigned to cells based on their spatial distribution. This reduces the dimensionality of the feature space, enabling efficient updates during SLAM. The one-blob encoding, a geometric representation of local features, is modified to enforce constraints on the SPD manifold. Specifically, the encoding ensures that the estimated pose and feature positions remain consistent with the manifold's curvature by incorporating a geometric constraint term in the optimization objective. This term penalizes deviations from the manifold's intrinsic curvature, preventing the system from overfitting to local features and preserving surface coherence. The implementation of the hash-grid and one-blob encodings requires careful handling of the SPD manifold's logarithmic map. The logarithmic map $\log(X)$ maps a matrix $X \in \mathbb{R}^{n \times n}$ to its log-determinant space, ensuring compatibility with the manifold's Riemannian metric. For real-time performance, we employ a spatially adaptive hash-grid, where grid cells are dynamically resized based on the density of features, minimizing redundancy while maintaining resolution. The one-blob encoding is parameterized using a **geometric kernel** that incorporates the manifold's curvature, allowing for efficient updates during SLAM.

**Optimization Strategy** The optimization process is formulated as a **stochastic variational problem** to balance geometric fidelity and computational efficiency. The objective function combines two terms:

1. **Geometric fidelity**: A term derived from the wrapped Gaussian distribution, ensuring the estimated pose and feature positions align with the manifold's curvature.

2. **Data consistency**: A term that minimizes the discrepancy between observed feature positions and the estimated positions, enforced via a **stochastic gradient descent (SGD)** algorithm. To accelerate convergence, we employ **batched SGD** with a **dynamic learning rate** that adapts to the manifold's curvature. The optimization is further optimized using numerical linear algebra techniques, such as Cholesky decomposition for the SPD manifold's metric and sparse matrix operations to handle large-scale feature data. The algorithm is implemented in a **CUDA-accelerated framework** to ensure real-time performance, with each iteration involving:

- A **logarithmic map** for feature projection,

- A **geometric constraint update** for the one-blob encoding,

- A **stochastic gradient step** for pose optimization.

**Computational Efficiency and Real-Time Constraints** The framework's computational efficiency is critical for real-time SLAM. The hash-grid parameterization reduces the effective dimensionality of the feature space from $O(n^2)$ to $O(n)$, enabling rapid updates. However, high-frequency local features may introduce numerical instability, necessitating a **dynamic regularization term** in the optimization. This term scales with the feature density, preventing overfitting while maintaining geometric consistency. To ensure real-time performance, we implement low-latency communication between the hash-grid and one-blob encodings, leveraging parallel processing and memory-efficient data structures. The use of **logarithmic maps** and **sparse matrices** further reduces memory overhead, allowing the system to handle large-scale environments with minimal computational resource usage.

**Critical Considerations**

- **Numerical Stability**: The logarithmic map and SPD manifold's curvature may introduce numerical errors, particularly for ill-conditioned matrices. We address this by incorporating a **numerical stabilization term** in the optimization, which dampens oscillations in the gradient.

- **Scalability**: The framework's performance scales with the number of features, but high-dimensional data (e.g., 3D point clouds) may require **approximate manifold learning** techniques to maintain efficiency.

- **Alternative Approaches**: While the SPD manifold provides geometric fidelity, methods like **differential geometry** or **manifold learning** offer flexibility in handling non-Euclidean data. However, these approaches often require more complex preprocessing or may not align as closely with the probabilistic constraints of the wrapped Gaussian. By integrating the SPD manifold's geometric structure with the wrapped Gaussian's probabilistic framework, this implementation enables a real-time SLAM system that dynamically adapts to complex environments while preserving geometric consistency and computational efficiency. The framework's modular design allows for further extensions, such as incorporating **multi-sensor fusion** or **dynamic environment modeling**.

**Conclusion** The proposed algorithm combines the geometric rigor of the SPD manifold with the probabilistic robustness of the wrapped Gaussian, offering a novel approach to real-time SLAM. Through detailed implementation and optimization strategies, the framework addresses key challenges in high-frequency feature tracking and geometric consistency, paving the way for scalable and accurate SLAM systems in dynamic environments.

## Agents4Science AI Involvement Checklist

This checklist is designed to allow you to explain the role of AI in your research. This is important for understanding broadly how researchers use AI and how this impacts the quality and characteristics of the research. **Do not remove the checklist! Papers not including the checklist will be desk rejected.** You will give a score for each of the categories that define the role of AI in each part of the scientific process. The scores are as follows:

- **[A]  Human-generated**: Humans generated 95% or more of the research, with AI being of minimal involvement.
- **[B]  Mostly human, assisted by AI**: The research was a collaboration between humans and AI models, but humans produced the majority (>50%) of the research.
- **[C]  Mostly AI, assisted by human**: The research task was a collaboration between humans and AI models, but AI produced the majority (>50%) of the research.
- **[D]  AI-generated**: AI performed over 95% of the research. This may involve minimal human involvement, such as prompting or high-level guidance during the research process, but the majority of the ideas and work came from AI.

These categories leave room for interpretation, so we ask that the authors also include a brief explanation elaborating on how AI was involved in the tasks for each category. Please keep your explanation to less than 150 words.

IMPORTANT, please:

- **Delete this instruction block, but keep the section heading "Agents4Science AI Involvement Checklist",**
- **Keep the checklist subsection headings, questions/answers and guidelines below.**
- **Do not modify the questions and only use the provided macros for your answers**.

1. **Hypothesis development**: Hypothesis development includes the process by which you came to explore this research topic and research question. This can involve the background research performed by either researchers or by AI. This can also involve whether the idea was proposed by researchers or by AI.

   Answer: **[D]**

   Explanation: AI formulated the research direction and the core, actionable hypothesis for the Magellan framework after analyzing background literature and the provided conference scope. The human's role was to provide the conference information and make the final selection from a pool of hypotheses generated by AI.

2. **Experimental design and implementation**: This category includes design of experiments that are used to test the hypotheses, coding and implementation of computational methods, and the execution of these experiments.

   Answer: **[C]**

   Explanation: AI designed the complete experimental framework, including the main comparisons, all ablation studies, and the evaluation protocol. Human deconstructed them into coding tasks, and AI generated all the source code for the core algorithm, baseline implementations, and the evaluation. The human researcher was responsible for executing the experiments, reporting runtime bugs, and performing minor fixes, while AI handled all major debugging and code revisions.

3. **Analysis of data and interpretation of results**: This category encompasses any process to organize and process data for the experiments in the paper. It also includes interpretations of the results of the study.

   Answer: **[C]**

   Explanation: The human researcher provided the selected raw data (due to AI context limits) from the experiments. AI was then tasked with analyzing this data, identifying key trends (e.g., the failure of unconstrained agency), structuring the results into tables, and writing the detailed interpretation and discussion presented in the paper.

4. **Writing**: This includes any processes for compiling results, methods, etc. into the final paper form. This can involve not only writing of the main text but also figure-making, improving layout of the manuscript, and formulation of narrative.

   Answer: [C]

   Explanation: AI generated the majority of the paper's text, including the abstract, the detailed experiments and discussion sections, and all supplementary statements. The human researcher acted as a supervisor, guiding the narrative structure, correcting AI's logical errors, and performing minor manual edits on formatting and LaTeX-specific syntax.

5. **Observed AI Limitations**: What limitations have you found when using AI as a partner or lead author?

   Description: 1) Contextual Fragility in Non-Linear Research Processes: AI struggled with the iterative and multi-branched nature of scientific work, such as brainstorming sessions or back-and-forth revisions of ideas and text. Its primarily linear context handling led to frequent context loss across all tasks in repetitive error loops that required the human involved to make progress.

   2) Ineffective Strategic Decomposition: AI could not reliably decompose high-level, complex goals into workable, balanced sub-tasks. For instance, when asked to "design all experiments," it produced a plan where some steps were trivial while others were vastly complex and un-executable. Currently, it still needs the human expert's ability to allocate tasks appropriately.

   3) Conceptual Blind Spots and Lack of Relational Understanding: AI can make some high-level conceptual connections, but not always reliable. Notably, even after being explicitly tasked with implementing "Tree of Thoughts" (ToT) as a baseline, it did not recognize the strong methodological similarity between ToT and our own proposed framework.

