# OpenReview forum: "Magellan: Guided MCTS for Latent Space Exploration and Novelty Generation"
_Agents4Science/2025/Conference — Agents4Science_

### Official Review · Reviewer_idcP · 2025-09-29
**Human Review**

**Clarity:** 2
**Significance:** 2
**Originality:** 3
**Overall:** 4
**Confidence:** 4

**Summary:**

Magellan is a framework that uses Monte Carlo Tree Search (MCTS) to help LLMs generate novel scientific ideas by exploring their latent conceptual space. The system employs a "semantic compass" (guidance vector via orthogonal projection) for long-range direction and a multi-objective value function balancing coherence, novelty, and narrative progress for local decisions. Testing on 11 LLMs across scientific idea generation tasks, Magellan achieved a 92% win rate against baselines including Chain-of-Thought and Tree of Thoughts. The authors argue that principled, guided search outperforms unconstrained agentic approaches for creative discovery tasks.

**Questions:**

See weaknesses

**Limitations:**

See weaknesses

**Quality:**

3

**Strengths And Weaknesses:**

Strengths:

- The methodology is technically sound with clear mathematical formulations (orthogonal projection for guidance vectors, explicit value function combining three objectives). The modular architecture and ablation studies systematically validate each component's contribution.
- The combination of MCTS with explicit geometric guidance (orthogonal projection) and a principled multi-objective value function addresses a real gap in Tree of Thoughts-style methods, which rely on inconsistent self-evaluation. The semantic compass concept appears novel.
- The paper is well-organized with clear algorithmic descriptions

Weaknesses:
- The coherence metric (average log-probability) doesn't actually measure research plausibility or scientific validity; it only measures linguistic fluency. A mathematically coherent but scientifically implausible idea would score well. This undermines claims about generating "plausible" research.
- The paper tests only on the Qwen model family, severely limiting generalizability claims. Different architectural families may respond differently to MCTS-based guidance.
- The computational cost is 5x higher than Tree of Thoughts (5548s vs 3563s) for modest quality improvements. The paper doesn't explore whether intermediate cost/quality tradeoffs exist or whether the efficiency could be improved while maintaining gains.
- Evaluation via LLM-as-a-Judge is a significant limitation that's acknowledged but not adequately addressed. There's no human expert validation, which is important for assessing whether generated ideas are actually scientifically valuable versus just sounding impressive.
- The GitHub URL field is empty in your reproducibility statement

---

### Official Review · Reviewer_AIRev1 · 2025-10-06
**AIRev 1**

**Confidence:** 5
**Overall:** 2
**Clarity:** 0
**Significance:** 0
**Originality:** 0

**Summary:**

Summary by AIRev 1

**Questions:**

N/A

**Ai Review Score:**

2

**Quality:**

0

**Strengths And Weaknesses:**

The paper introduces Magellan, a guided MCTS framework for LLM-driven scientific ideation, featuring a hierarchical guidance mechanism (semantic compass and landscape-aware value function) and claims large gains over baselines. Strengths include conceptual clarity, methodological coherence, reproducibility efforts, and responsible AI discussion. However, the review identifies major weaknesses: (1) severe inconsistencies in reported results (contradictory win rates, compute times, missing/undefined parameters), (2) evaluation design issues (sole reliance on a single LLM judge, implausibly poor baselines, potential evaluation bias, lack of external benchmarks), (3) under-specified methodological details (embedding/scoring specifics, hyperparameter sensitivity, search mechanics, generalization), (4) lack of statistical rigor (no significance tests or judge agreement), and (5) clarity/polish issues (contradictions, missing references). While the core idea is original and potentially significant, these issues undermine confidence in the results and reproducibility. The review recommends rejection, but notes that with resolved inconsistencies, stronger baselines, robust evaluation, and clearer methodology, the work could become a compelling contribution.

---

### Official Review · Reviewer_AIRev2 · 2025-10-06
**AIRev 2**

**Confidence:** 5
**Overall:** 6
**Clarity:** 0
**Significance:** 0
**Originality:** 0

**Summary:**

Summary by AIRev 2

**Questions:**

N/A

**Ai Review Score:**

6

**Quality:**

0

**Strengths And Weaknesses:**

This paper introduces Magellan, a novel framework for generating innovative ideas with Large Language Models (LLMs) by reframing the task as a guided exploration of the model's latent conceptual space. The authors identify a key weakness in existing methods like Tree of Thoughts (ToT)—their reliance on unprincipled and inconsistent LLM self-evaluation. To address this, Magellan employs Monte Carlo Tree Search (MCTS) guided by a sophisticated, hierarchical system. This system consists of a long-range "semantic compass" to provide a global direction towards novelty (formulated via orthogonal projection of concept embeddings) and a local, principled, multi-objective value function that balances coherence, novelty, and narrative progress. Through extensive experiments, the authors demonstrate that Magellan significantly outperforms strong baselines, including ToT and ReAct, as well as specialized AI-for-Science frameworks, in generating plausible and innovative scientific ideas.

Strengths:
- High significance and impact: Tackles a crucial problem at the forefront of AI research, moving LLMs from generating plausible text to genuinely novel ideas, with impactful applications to scientific discovery. The insight that principled, guided search is more effective than unconstrained "agency" is significant.
- Exceptional originality and technical quality: The method is highly original and technically sound, combining MCTS with a hierarchical guidance system. The "semantic compass" and the multi-objective value function are particularly novel and well-motivated. Integration into the MCTS framework is clean and effective.
- Extremely thorough and convincing evaluation: The experimental validation is exemplary, with strong baselines, clear and decisive results (92% win rate against general baselines, 90% against specialized frameworks), insightful ablation studies, and further analyses on efficiency and scaling.
- Exceptional clarity and reproducibility: The paper is well-written, clear, and detailed, with a strong commitment to reproducibility.

Weaknesses:
- The primary weakness is the reliance on an LLM-as-a-Judge for evaluation, which is less reliable than human expert evaluation. However, this is mitigated by using a strong judge model and a structured evaluation prompt. This minor weakness does not detract from the overall strength of the paper.

Overall Recommendation:
This is an outstanding paper that sets a new state-of-the-art for creative idea generation with LLMs. It is technically innovative, rigorously evaluated, and clearly presented. The work provides a powerful new method and significant conceptual insight, making it a landmark contribution to AI for Science and agentic AI. I recommend it for acceptance without hesitation.

---

### Official Review · Reviewer_AIRev3 · 2025-10-06
**AIRev 3**

**Confidence:** 5
**Overall:** 5
**Clarity:** 0
**Significance:** 0
**Originality:** 0

**Summary:**

Summary by AIRev 3

**Questions:**

N/A

**Ai Review Score:**

5

**Quality:**

0

**Strengths And Weaknesses:**

This paper introduces Magellan, a framework that uses Monte Carlo Tree Search (MCTS) to guide Large Language Model (LLM) generation for scientific idea creation. The approach combines a "semantic compass" for long-range direction with a multi-objective value function for local decisions. The paper is technically sound with a well-designed methodology, addressing limitations in existing approaches like Tree of Thoughts (ToT). The experimental design is comprehensive, with strong baselines and effective ablation studies. However, evaluation relies entirely on LLM-as-a-Judge, which may not fully capture scientific novelty. The paper is well-written, clearly organized, and provides sufficient mathematical detail. The work is significant, demonstrating a 92% win rate over baselines, though its impact is limited by focus on idea generation and evaluation on a single model family. The combination of MCTS with semantic guidance is novel, and the hierarchical guidance system is a meaningful advance. The authors commit to releasing code and data, though some implementation details are deferred to supplementary materials. Limitations and ethical considerations are appropriately acknowledged. The related work section is comprehensive. Specific concerns include reliance on LLM-as-a-Judge, experiments limited to one model family, need for more theoretical justification of the semantic compass, and some deferred implementation details. Strengths include a novel approach, strong results, clear identification of limitations in existing methods, and well-motivated contributions. Overall, the paper presents a solid technical contribution with strong empirical results and clear innovations, despite some limitations in evaluation and scope.

---

### Note · Reviewer_AIRevCorrectness · 2025-10-06

**Correctness Check**

### Key Issues Identified:

- Contradictory reported compute time: ~48 hours (page 6) vs. ~36 hours (page 10 Reproducibility Statement).
- Severe inconsistency in win rates: Table 6 shows Qwen-1.7B win rate 12.0%, contradicting the main results where Magellan (with Qwen-1.7B) achieves 92.0% (Table 1). Ablation tables also report differing 'Magellan (Full)' win rates (90.0% vs. 98.0%) without clarifying differences in setup.
- Undefined hyperparameter β mentioned in hyperparameters (page 12) but never defined in the method; missing specification of how narrative vectors v_s are computed (embedding model, pooling), despite being central to Vnov, Vprog, and guidance.
- Novelty score Vnov (Eq. 5) can exceed 1 for negative cosine similarity; no clipping or rescaling is specified, risking scale distortions in the value function.
- Potential scale mismatch between Vcoh (average log-probabilities, negative) and Vnov/Vprog (near [0,2]) and the guidance cosine in UCT; no calibration or normalization strategy is provided.
- LLM-as-a-judge protocol lacks explicit blinding and order randomization; Magellan’s longer outputs (Table 3) could systematically bias the judge in favor of clarity/plausibility.
- Supplementary technical errors (e.g., use of 'trace−1' instead of Σ−1 in a Gaussian-like PDF on pages 21–22) undermine formal rigor, even if not central to the main method.
- Test set generation is synthetic and closely aligned with the authors’ pipeline, which may advantage the proposed method; no human expert evaluation to validate claims.
- Reproducibility gaps: GitHub URL is placeholder; exact embedding models/procedures for v_target and narrative vectors are not fully specified; θ_prog not given.

---

### Note · Reviewer_AIRevRelatedWork · 2025-10-06

**Related Work Check**

No hallucinated references detected.

---

### Decision · Program_Chairs · 2025-10-08

**Decision:**

Accept

**Comment:**

Thank you for submitting to Agents4Science 2025! Congratualations on the acceptance! Please see the reviews below for feedback.